# Contextual Similarity Aggregation with Self-attention for Visual Re-ranking

**Jianbo Ouyang**[1*]  **Hui Wu**[1*]  **Min Wang**[2†]  **Wengang Zhou**[1,2†]  **Houqiang Li**[1,2]

[1]CAS Key Laboratory of Technology in GIPAS, EEIS Department
University of Science and Technology of China
[2]Institute of Artificial Intelligence, Hefei Comprehensive National Science Center
`{ouyjb,wh241300}@mail.ustc.edu.cn`
`wangmin@iai.ustc.edu.cn, {zhwg,lihq}@ustc.edu.cn`

## Abstract

In content-based image retrieval, the first-round retrieval result by simple visual feature comparison may be unsatisfactory, which can be refined by visual re-ranking techniques. In image retrieval, it is observed that the contextual similarity among the top-ranked images is an important clue to distinguish the semantic relevance. Inspired by this observation, in this paper, we propose a visual re-ranking method by contextual similarity aggregation with self-attention. In our approach, for each image in the top-$K$ ranking list, we represent it into an affinity feature vector by comparing it with a set of anchor images. Then, the affinity features of the top-$K$ images are refined by aggregating the contextual information with a transformer encoder. Finally, the affinity features are used to recalculate the similarity scores between the query and the top-$K$ images for re-ranking of the latter. To further improve the robustness of our re-ranking model and enhance the performance of our method, a new data augmentation scheme is designed. Since our re-ranking model is not directly involved with the visual feature used in the initial retrieval, it is ready to be applied to retrieval result lists obtained from various retrieval algorithms. We conduct comprehensive experiments on four benchmark datasets to demonstrate the generality and effectiveness of our proposed visual re-ranking method.

## 1  Introduction

In instance image retrieval, the goal is to efficiently identify images containing the same object or describing the same scene with the query image from a large corpus of images. Towards this goal, many works have emerged in recent years [42, 21, 36, 26, 24]. With the development of deep learning, a great number of methods leverage convolutional neural network (CNN) as feature extractor [38, 3, 44, 4, 14, 34, 25, 35, 45] to replace or combine with the classic SIFT feature [23]. Generally, the performance of the initial retrieval results by simple comparison of visual feature may not be satisfactory. To refine it, many visual re-ranking techniques have been proposed [9, 19, 48].

Popular visual re-ranking techniques include query expansion, geometric context verification, kNN-based re-ranking, and diffusion-based methods, *etc*. Query Expansion (QE) [9, 10, 39, 34, 30] computes an average feature of the query and the top-ranked images to update the original query

---

[*]The first and second authors contribute equally to this work.

[†]Corresponding authors: Min Wang and Wengang Zhou

  Code will be released at `https://github.com/MCC-WH/CSA`.

feature, which is then used for a second-round retrieval. Geometric Context Verification (GCV) [30, 12, 20, 2] leverages the geometric context of local features to remove the false matches in the original results. Besides, if two images belong to the $k$-reciprocal nearest neighbors ($k$NN) of each other, these two images have a high probability of being relevant. Based on this observation, lots of $k$-Nearest Neighbors ($k$NN) based re-ranking methods thus emerge and achieve outstanding performance [19, 48, 50]. Diffusion-based methods [51, 11, 18] consider the global similarity of all images and perform similarity propagation iteratively. Most of the above methods do not involve training, thus can be quickly equipped with various features. The recent learning-based methods such as LAttQE [16] and GSS [22], achieve better performance but require training a specific model for each kind of feature.

Different from the methods mentioned above, in this paper, we propose a novel visual re-ranking method by contextual information aggregation. The initial retrieval results generated by feature comparison only consider pairwise image similarity, but ignore the rich contextual information contained in the ranking list. As illustrated in [29], if two images are mutually relevant, they share similar distances from a set of anchor images. Thus, we directly select the top-$L$ retrieval results as anchors for each query to fully model the contextual information in the ranking list. We define an affinity feature for each image in top-$K$ candidates by computing the similarity between it and the anchor images. To further promote the re-ranking quality, we propose a new data augmentation method.

Moreover, inspired by Query Expansion (QE) which employs the information of top-ranked images to update the query feature, we propose to update affinity features of each top-$K$ candidates by aggregating the affinity features of other candidates to promote the re-ranking performance with transformer encoder [46]. For each image in the top-$K$ candidates, our re-ranking model dynamically aggregates the affinity features of other candidates based on the importance between candidates by computing their similarities using the affinity features. The output of our re-ranking model can be regarded as the refined affinity features for top-$K$ candidates which contain more contextual information.

To train the re-ranking model, two loss functions are introduced. Firstly, we use a contrastive loss to restrain the updated affinity features so that the relevant images have large similarity, and vice versa. Besides, we exploit a Mean Squared Error (MSE) loss to reserve the information in original affinity features by restricting the difference between the original affinity features and the refined affinity features. During the inference time, we compute the affinity features for the top-$K$ candidates and utilize the transformer encoder to refine the features. Then we re-rank these candidates by computing the cosine similarity between the refined affinity features. The rank of images outside the top-$K$ ranking list remains unchanged.

Note that our re-ranking model is not directly involved with the original visual feature. Instead, it computes the affinity features for top-$K$ images, which serve as the input in our method. Therefore, it can be combined with various existing image retrieval algorithms including representation learning and re-ranking methods, and further enhance the retrieval performance with low computational overhead. We conduct comprehensive experiments on four benchmark datasets to prove the generality and effectiveness of our proposed visual re-ranking method. Besides, the time and memory complexity of our re-ranking method is lower than the state-of-the-art re-ranking methods.

## 2   Related Work

In this section, we review the related works on visual re-ranking including query expansion, geometric context verification, $k$NN based methods, and diffusion-based methods.

**Query Expansion.** Query Expansion (QE) uses the information of the top-ranked images obtained from the initial retrieval to refine the representation of the original query for further retrieval. Average query expansion (AQE) [9] aggregates the features of top-$K$ retrieval results by average pooling. Average query expansion with decay (AQEwD) [15] proposes a weighted average aggregation, where the weights decay over the rank of retrieved images. Discriminative query expansion (DQE) [1] regards the top-ranked images as positive images, and regards the bottom-ranked images as negative images to train a linear SVM. Then DQE sorts images according to their signed distance from the decision boundary. In [34], $\alpha$-weighted query expansion ($\alpha$QE) weights the top-$K$ images by their

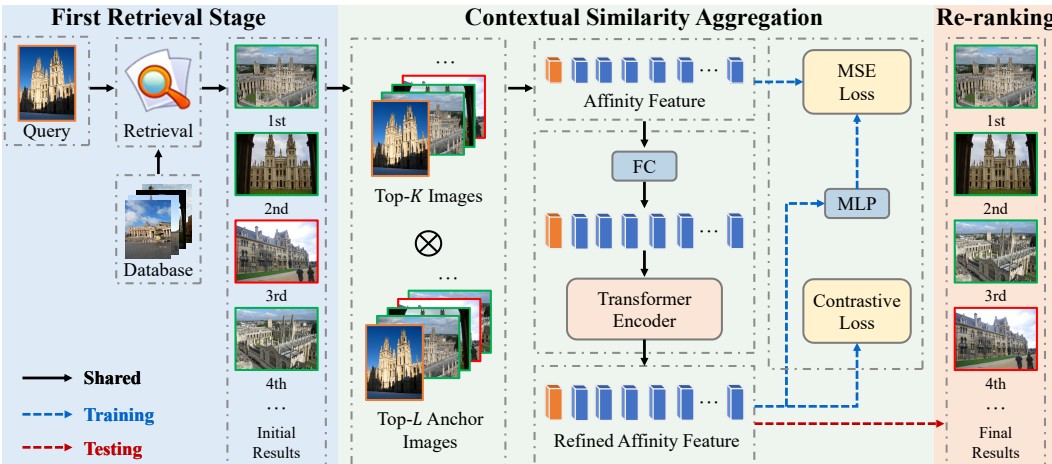

Figure 1: The pipeline of our method. In the first retrieval stage marked in blue, given a query, we perform the first-round retrieval to get the initial ranking list. In the second stage marked in green, we first obtain the affinity features for the query (orange cuboid) and top-$K$ candidates (blue cuboids) with top $L$ images as anchor images. The transformer encoder is used to refine the affinity features. The network is trained by minimizing the contrastive loss and MSE loss using the refined affinity features. In the re-ranking phase marked in orange, we recalculate the similarity scores between the query and the top $K$ candidates using the refined affinity features. Best viewed in color.

cosine similarity with the query. LAttQE [16] proposes an attention-based model to learn the weights of aggregation.

**Geometric Context Verification.** Geometric Context Verification uses the geometric context of local features to remove the false matches. Some approaches [9, 30, 37] estimate the transformation model to verify the local correspondences by RANSAC-based methods. RANSAC [12] generates hypotheses on random sets of correspondences and then identifies a geometric model with maximum inliers. Spatial coding [52] proposes to represent the relative spatial locations among local features into binary maps for efficient local matching verification. DSM[40] leverages the MSER regions which are detected on the activation maps of the convolutional layer. Then DSM regards these regions as local features for spatial verification.

$k$-**NN-based Re-ranking.** The $k$-reciprocal nearest neighbors of an image are considered as highly relevant candidates [49, 32, 39]. Contextual Dissimilarity Measure (CDM) [19] iteratively regularizes the average distance of each point to its neighbors to update the similarity matrix. Visual Rank [48] determines the centrality of nodes on a similarity graph according to the link structures among images to rank images. In CRL [27], a lightweight CNN model is trained to explore the contextual information and learn the relevance between images. GSS [22] proposes an unsupervised method for re-ranking based on graph neural network neighbor encoding and pairwise similarity separation loss. In [50], the $k$-reciprocal feature is designed by encoding the $k$-reciprocal nearest neighbors into a single vector to perform re-ranking.

**Diffusion-based Re-ranking.** Diffusion technique is initially designed for ranking on manifolds [51], and it has been successfully applied to many computer vision tasks, such as image classification, object detection, image retrieval, *etc*. In image retrieval, a lot of research concentrates on propagating similarity through the $k$NN graph [11]. Diffusion is usually used as a re-ranking method, and it has achieved state-of-the-art performance on many benchmarks [28, 47, 5, 18, 33, 8, 7, 6, 13, 28]. Diffusion considers the underlying manifold structure and explores the internal relationships between images. As one of the representative methods, DFS [18] proposes a regional diffusion mechanism to further improve the recall of small objects based on the neighborhood graphs. Then it employs conjugate gradient method to compute the closed-form solution of diffusion as well as maintaining time efficiency and accuracy.

## 3 Method

### 3.1 Framework Overview

In image retrieval, the database is defined as $D = \{I_1, I_2, \cdots, I_N\}$, where $I_i$ denotes the $i$th database image and $N$ is the database size. Usually, images are first mapped to high-dimensional vectors

through a feature extractor $\phi(\cdot)$. We define the feature representation of images as $\boldsymbol{f} = \phi(I) \in R^d$. Given a query, we obtain an initial ranking list of the top-$K$ images $R = [r_1, r_2, \cdots, r_K]$ by a retrieval algorithm, where $r_i$ denotes the ID of the $i$th image. For convenience of the following discussion, we assume the query image is returned at the first position in the ranking list, otherwise we directly insert it in the front of the ranking list. The features of top-$K$ images in the ranking list are described as $F_K = [\boldsymbol{f}_{r_1}, \boldsymbol{f}_{r_2}, \cdots, \boldsymbol{f}_{r_K}] \in R^{d \times K}$.

Our framework is shown in Fig. 1. With the initial retrieval ranking list obtained by a query, we propose to represent each of the top-$K$ candidates into an affinity feature vector by computing the cosine similarity between it and a set of anchor images. Then the affinity features are refined by a transformer encoder to aggregate the contextual information in the ranking list.

**Affinity Feature.** Generally, if two images are relevant to the query, they shall be relevant to each other [43]. Moreover, as revealed in [29], if two images are mutually relevant, their distances to a set of pre-defined anchor images shall be similar. Actually, it is a non-trivial issue to select a proper set of images as anchors. Generally, if the selected anchor images are far from the top-$K$ candidates in the ranking list, the variance of its distance with candidates is relatively small, which is difficult to provide useful information to distinguish the candidates. In other words, a better alternative is to choose the top-$L$ images in ranking list as the anchor images. With anchor images, we define the affinity features for the $i$th image in the rank list $R$ as follows:

$$\boldsymbol{a}_i = [\boldsymbol{f}_{r_i}^T \boldsymbol{f}_{r_1}, \boldsymbol{f}_{r_i}^T \boldsymbol{f}_{r_2}, \cdots, \boldsymbol{f}_{r_i}^T \boldsymbol{f}_{r_L}]^T, 1 \le i \le K. \tag{1}$$

After that, we take the affinity features of the $K$ images in $R$ as a sequence, which is fed into our re-ranking model for refinement. Note that the input of our proposed method is affinity feature sequence and is not directly related to the original feature, which endows our method with good generality.

**Contextual Similarity Aggregation with Transformer.** Although the affinity features have taken into account the contextual information in the ranking list, there may be inconsistency in the initial affinity features obtained by directly comparing with anchor images. In order to make the affinity features of related images more consistent, we propose a new re-ranking model by contextual information aggregation to dynamically refine the affinity features of different images. Specifically, with transformer encoder as the core component of our re-ranking model, we first map the affinity feature $\boldsymbol{a}_i \in R^L$ for the $i$th image to $\boldsymbol{a}' \in R^{L'}$ using a learnable projection matrix $W_p \in R^{L \times L'}$, and then feed the sequence $\boldsymbol{a}'_i, i = 1, 2, \cdots, K$ into a transformer encoder consisting of Multi-Head Attention (MHA) and Feed-Forward Networks (FFNs).

In MHA, each sequence element is updated by a weighted sum of all other elements based on the scaled dot-product similarity. The scaled dot-product attention first maps affinity features $\boldsymbol{a}'_i \, (i = 1, 2, \cdots, K)$ to Queries ($\boldsymbol{Q} \in R^{K \times d_s}$), Keys ($\boldsymbol{K} \in R^{K \times d_s}$) and Values ($\boldsymbol{V} \in R^{K \times d_s}$) with three learnable projection matrices. After that, the similarity between $\boldsymbol{Q}$ and $\boldsymbol{K}$ is aggregated with $\boldsymbol{V}$ together,

$$\text{Attention}(\boldsymbol{Q}, \boldsymbol{K}, \boldsymbol{V}) = \text{Softmax}(\frac{\boldsymbol{Q}\boldsymbol{K}^T}{\sqrt{d_s}})\boldsymbol{V}, \tag{2}$$

which can be seen as using the similarity between images as weights to aggregate affinity features. The multi-head structure concatenates and fuses the outputs of multiple scaled dot-product attention modules using the learnable projection $\boldsymbol{W}_M$. MHA is defined as follows:

$$\text{MHA}(\boldsymbol{Q}, \boldsymbol{K}, \boldsymbol{V}) = \text{Concatenation}(h_1, h_2, \cdots, h_{N_H})\boldsymbol{W}_M, \tag{3}$$

where

$$h_i = \text{Attention}(\boldsymbol{Q}_i, \boldsymbol{K}_i, \boldsymbol{V}_i), i = 1, 2, \cdots, N_H, \tag{4}$$

and $N_H$ is the head number. Note that to update the affinity feature of an image, the affinity features of all the relevant candidates should be considered as contextual information. A single-head self-attention layer limits the ability of the model to focus on one or more relevant candidates simultaneously without affecting other equally important candidates. This can be achieved by projecting the original affinity features into different representation subspaces. Specifically, in MHA, different projection matrices for $\boldsymbol{Q}$, $\boldsymbol{K}$, and $\boldsymbol{V}$ are used for different heads, and these matrices can project affinity features into different subspaces. The output is normalized via Layer Normalization (LN) and finally added to $\boldsymbol{S}'$ to form a residue connection,

$$\boldsymbol{S}' = \boldsymbol{S}' + \text{LN}(\text{MHA}(\boldsymbol{Q}, \boldsymbol{K}, \boldsymbol{V})). \tag{5}$$

In addition to MHA, the transformer encoder layer has a position-wise feed-forward network.

We stack $n$ transformer encoder layers to obtain more consistent affinity features. The refined affinity features $\boldsymbol{y}_i, i = 1, 2, \cdots, K$ come from the output of the last layer of our re-ranking model.

**Data Augmentation.** According to the description above, we represent the top-$K$ images into affinity features, which serve as the input to our model. To improve the robustness of the re-ranking model and avoid overfitting, it is necessary to increase the size of the training sets. To this end, we design a data augmentation scheme oriented to our framework.

Considering that our training data comes from retrieval results, we propose to use different visual features to obtain the ranking lists of the same query. With different types of features to create the affinity features, the network will learn more information about the contextual information. For a query, we employ $M$ different visual feature extractors $\{\phi(\cdot)_1, \phi(\cdot)_2, \cdots, \phi(\cdot)_M\}$ to obtain different ranking lists, and compute their corresponding affinity features according to the previous section. The obtained $M$ affinity feature sequences for the same query are used as training samples. In this way, we enlarge the training set size to $M$ times of the original size and improve the robustness of our model.

## 3.2 Objective Functions

**Contrastive Loss.** The relevant images should have larger cosine similarity and vice versa, thus we design a contrastive loss for representation learning, which is defined as follows:

$$\mathcal{L}_C = -log \frac{\sum_{i=2}^{K} exp(sim(\boldsymbol{y}_1, \boldsymbol{y}_i)/\tau) \cdot \mathbb{1}(r_1 \text{ and } r_i \text{ are relevant})}{\sum_{i=2}^{K} exp(sim(\boldsymbol{y}_1, \boldsymbol{y}_i)/\tau)}, \tag{6}$$

with the cosine similarity:

$$sim(\boldsymbol{y}_i, \boldsymbol{y}_j) = \boldsymbol{y}_i^T \boldsymbol{y}_j / (\|\boldsymbol{y}_i\| \cdot \|\boldsymbol{y}_j\|), \tag{7}$$

where $\|\cdot\|$ is $L_2$ norm, $\mathbb{1}(\cdot)$ is the indicator function, $\tau$ is a temperature hyper-parameter, and $\boldsymbol{y}_1$ is the refined affinity feature of the query because we assume that query is on the top of the ranking list. The output affinity features of other images in ranking list are represented as $\boldsymbol{y}_i, 2 \leq i \leq K$. In the numerator, the sum is over relevant images, and in the denominator, the sum is over all top-$K$ images in the ranking list. Contrastive loss is minimized when query is similar to its relevant images and dissimilar to all other irrelevant images.

**MSE Loss.** In order to preserve the information contained in the original affinity features, we propose to use a Mean Squared Error (MSE) loss between the original affinity features and the refined affinity features. MSE loss is defined as follows:

$$\mathcal{L}_M = \sum_{i=1}^{K} \|\boldsymbol{a}_i - \text{MLP}(\boldsymbol{y}_i)\|^2, \tag{8}$$

where $\boldsymbol{s}_i$ and $\boldsymbol{y}_i$ is the original and refined affinity feature for the $i$th candidate, respectively. The Multilayer Perceptron (MLP) containing two layers with GELU non-linearity projects the refined affinity features back to the original affinity feature space to compute the MSE loss.

Combining Eq. (6) and Eq. (8), we define the final objective function of the proposed method as follows:

$$\mathcal{L} = \mathcal{L}_C + \lambda \mathcal{L}_M, \tag{9}$$

where $\lambda$ is a hyper-parameter to indicate the importance of MSE loss.

## 3.3 Re-ranking Processing

To aggregate the contextual information, we represent the top-$K$ images into affinity features and refine them by the transformer encoder. Because of the assumption that the query is at the first position of the ranking list, we compute its similarity with the $i$th image in ranking list in $R$ using the refined affinity features as follows:

$$s_i' = sim(\boldsymbol{y}_1, \boldsymbol{y}_i), i = 1, 2, \cdots, K. \tag{10}$$

Finally, all the $K$ images in the initial ranking list $R$ are re-ranked by their new similarity score defined in Eq. (10) in descending order. The order of the remaining images which is out of top-$K$ remains unchanged.

Table 1: The re-ranking latency and retrieval accuracy of our method at different re-ranking lengths $K$ on $\mathcal{R}$Oxf and $\mathcal{R}$Par datasets. The initial retrieval is denoted as R-GeM, which is the baseline in our experiments. Anchor images list length $L = 512$.

| Method | Re-ranking latency (ms) | Medium | | Hard | |
|---|---|---|---|---|---|
| | | $\mathcal{R}$**Oxf** | $\mathcal{R}$**Par** | $\mathcal{R}$**Oxf** | $\mathcal{R}$**Par** |
| R-GeM [34] | 0.0 | 67.3 | 80.6 | 44.3 | 61.5 |
| R-GeM+Ours($K$=128) | 7 | 72.7 | 82.0 | 50.0 | 64.3 |
| R-GeM+Ours($K$=256) | 11 | 75.1 | 83.7 | 53.4 | 67.7 |
| R-GeM+Ours($K$=512) | 19 | 77.0 | 85.6 | 57.0 | 71.3 |
| R-GeM+Ours($K$=768) | 28 | 77.8 | 86.9 | 58.3 | 73.4 |
| R-GeM+Ours($K$=1024) | 37 | 77.9 | 87.2 | 58.4 | 74.4 |
| R-GeM+Ours($K$=1280) | 46 | **78.3** | 87.5 | 59.0 | 75.0 |
| R-GeM+Ours($K$=1536) | 55 | 78.2 | **88.2** | **59.1** | **75.3** |

Table 2: The retrieval accuracy of our method at different anchor images list length $L$ are compared with the initial retrieval performance on $\mathcal{R}$Oxf and $\mathcal{R}$Par datasets. R-GeM denotes the first-round retrieval performance, which serves as the baseline in our experiments. Re-ranking length $K = 1024$.

| Method | Medium | | Hard | |
|---|---|---|---|---|
| | $\mathcal{R}$**Oxf** | $\mathcal{R}$**Par** | $\mathcal{R}$**Oxf** | $\mathcal{R}$**Par** |
| R-GeM [34] | 67.3 | 80.6 | 44.3 | 61.5 |
| R-GeM+Ours($L$=128) | 75.9 | 86.9 | 55.9 | 72.3 |
| R-GeM+Ours($L$=256) | 77.5 | 86.8 | 57.4 | 73.0 |
| R-GeM+Ours($L$=512) | **77.9** | 87.2 | **58.4** | **74.4** |
| R-GeM+Ours($L$=768) | 77.0 | 87.2 | 57.0 | 73.7 |
| R-GeM+Ours($L$=1024) | 76.5 | 87.1 | 56.1 | 73.4 |
| R-GeM+Ours($L$=1280) | 76.4 | **87.3** | 56.6 | 73.7 |

## 4 Experiment

### 4.1 Experiment setup

**Image Representation.** We extract global visual features for both the query and database images using a ResNet101 [17] backbone with GeM pooling [34]. The best-performing model fine-tuned on the GL18 [25] is used. The resulting feature is denoted as R-GeM. To verify the robustness of our method when the training and testing sets are represented by different features, we also utilize other four models to extract testing image features: the off-the-shelf version of ResNet101 with R-MAC pooling [44], the off-the-shelf version of ResNet101 with GeM pooling, the fine-tuned version of ResNet101 with Max pooling [44], and the fine-tuned version of VGG16 [41] with GeM pooling.

**Evaluation Datasets and Metrics.** Four image retrieval benchmark datasets, named Revisited Oxford5k ($\mathcal{R}$Oxf), Revisted Paris6k ($\mathcal{R}$Par), $\mathcal{R}$Oxf + $\mathcal{R}$1M, and $\mathcal{R}$Par + $\mathcal{R}$1M, are used to evaluate our method. The $\mathcal{R}$Oxf [33] and $\mathcal{R}$Par [33] datasets are the revisited version of the original Oxford5k [30] and Paris6k datasets [31]. These two datasets both contain 70 query images depicting buildings, and additionally include 4,993 and 6,322 database images, respectively. $\mathcal{R}$Oxf + $\mathcal{R}$1M and $\mathcal{R}$Par + $\mathcal{R}$1M are the large-scale versions of $\mathcal{R}$Oxf and $\mathcal{R}$Par which combine a set of 1M distractor images with the small ones. Mean Average Precision (mAP) [30] is used to evaluate the performance. We report the Medium and Hard performance of the four datasets mentioned above. The computation is performed on a single 2080Ti GPU.

**Training Details.** rSfM120k [34] is used to create training samples. It includes images selected from 3D reconstructions of landmarks and city scenes. In total, 91,642 images from 551 3D models are used for training. Each image in the training set is considered as a query image, and the others are database images. The first-round retrieval results of each query form a training sample. For each query image, the image with the same 3D reconstruction cluster id is considered as a positive sample and vice versa as a negative sample. We select the top-512 returned images for each query image to form a training sample and select the top-512 images as anchor images for computing the affinity features. To realize data augmentation, we extract the training image features using multiple models. Specifically, for each image in the training set, we extract features using fine-tuned Resnet50,

Table 3: mAP performance of the proposed model with data augmentation. Re-ranking length K=1024, anchor images list length $L = 512$. rSfM120k denotes the original training set. AugrSfM120k denotes the training set with data augmentation.

| Training Dataset | Training nums | Medium | | Hard | |
|---|---|---|---|---|---|
| | | $\mathcal{R}$Oxf | $\mathcal{R}$Par | $\mathcal{R}$Oxf | $\mathcal{R}$Par |
| rSfM120k [34] | 91,642 | 77.9 | 87.2 | 58.4 | 74.4 |
| AugrSfM120k | 274,926 | **78.5** | **88.1** | **60.0** | **76.3** |

Resnet101, and Resnet152 with GeM pooling, respectively, to construct multiple different affinity features. We denote the training set with data augmentation as AugrSfM120k.

Our model consists of a stack of 2 transformer encoder layers, each with 12 heads of 64 dimensions. The fully connected layers within the encoder layers have 4 times more dimensions than the hidden dimension. SGD is used to optimize the model, with an initial learning rate of 0.1, a weight decay of $10^{-5}$, and a momentum of 0.9. We use a cosine scheduler to gradually decay the learning rate to 0. The temperature in Eq. (6) is set as 2.0. The batch size is set to 256. The model is trained for 100 epochs on four 2080Ti GPUs.

## 4.2 Ablation Study

**Re-ranking Length.** In our method, we re-rank the initial top-$K$ images of the ranking list and keep the order of the remaining images unchanged. For training, we select the fixed-length top-512 results as training samples. Since our transformer encoder involves no position embedding and the input length of the model is variable, we can change the length of the re-ranking during testing. The re-ranking performance of different $K$ on $\mathcal{R}$Oxf and $\mathcal{R}$Par and the corresponding time required are shown in Table 1. The results show that the performance of re-ranking keeps improving as $K$ increases, and the performance starts to saturate after $K$ is larger than 1024. For practical applications, we can take a trade-off between accuracy and latency time of re-ranking.

Table 4: mAP performance of the proposed model with different feature types. Te re-ranking model is trained by fine-tuned R-GeM. Re-ranking length $K = 1024$. V: VGG16 [41]; R: ResNet101 [17]; [O]: Off-the-shelf networks pretrained on ImageNet; RMAC: regional max-pooling [44]; GeM: generalized-mean pooling [34]; MAC: max-pooling [44].

| Method | Training feature | Medium | | Hard | |
|---|---|---|---|---|---|
| | | $\mathcal{R}$Oxf | $\mathcal{R}$Par | $\mathcal{R}$Oxf | $\mathcal{R}$Par |
| R-RMAC[O] [44] | - | 51.2 | 74.0 | 21.4 | 51.7 |
| **R-RMAC[O]+Ours** | R-GeM | **56.8** | **81.9** | **30.5** | **65.7** |
| R-GeM[O] [34] | - | 50.3 | 73.0 | 23.0 | 50.9 |
| **R-GeM[O]+Ours** | R-GeM | **55.0** | **81.5** | **30.3** | **65.6** |
| R-MAC [44] | - | 63.3 | 76.6 | 35.7 | 55.5 |
| **R-MAC+Ours** | R-GeM | **73.2** | **86.0** | **52.8** | **72.1** |
| V-GeM [34] | - | 61.6 | 69.3 | 34.3 | 44.9 |
| **V-GeM+Ours** | R-GeM | **73.3** | **81.5** | **50.0** | **73.0** |

**Length of Anchor Images List.** Our method updates the affinity features to capture the contextual information in the ranking list, and the choice of anchor images plays a key role. We select the top-$L$ images of the returned list as the anchor images for each query. Table 2 shows the effect of different anchor image lengths. The re-ranking achieves a large improvement relative to the baseline for all settings. When $L$ is small, it limits the expressiveness of the affinity vector, and conversely when $L$ is large, the proportion of images associated with the query image is too small, introducing a lot of noisy anchor images and limiting the discriminative power of the model. When $L = 512$, the best or competitive performance is achieved on all datasets.

**Model Variants.** In Figure 2, we show the impact of different model variants on the retrieval performance. The optimal performance is achieved when the transformer depth is equal to 2 and the hidden layer dimension is 1024. As seen in Figure 2(a), when the depth is zero, the model contains no

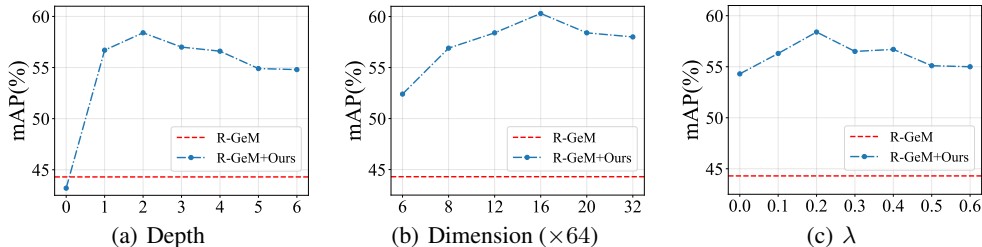

Figure 2: Impact of transformer model variants and the weight of the MSE loss on mAP on $\mathcal{R}$Oxf with Hard evaluation protocols. All model variants are evaluated with re-ranking length $K = 1024$ and anchor image list length $L = 512$. (a) Comparisons of different transformer depth $n$. The hidden size $L' = 768$, and head number $N_h = 12$. (b) Impact of the hidden size $L'$. Transformer depth is kept as 2. (c) Impact of the weight of the MSE loss. The model takes the default settings.

self-attention module and is simply a stack of fully connected layers, at which point the performance is lower than the baseline features, highlighting how important the self-attention is for our model. Figure 2(b) shows that the model performance rises and then falls as the model capacity increases. This can be attributed to the overfitting of the model.

**MSE Loss.** The effect of the MSE loss on the model performance is shown in Figure 2(c). It can be seen that as $\lambda$ increases, the performance gradually rises and reaches the highest when $\lambda = 0.2$. An excessively large $\lambda$ makes the model learning be dominated by the MSE loss and the performance decreases.

**Dataset augmentation.** As shown in Table 3, when using data augmentation, we can increase the training set to three times the original size, achieving higher performance relative to the baseline for the model trained without data augmentation. The re-ranking setting is the same as the compared methods which uses ResNet101-GeM feature to perform retrieval and re-rank using our trained model.

### 4.3 Cross Feature Testing

Our method just requires the affinity feature generated by comparing the image with anchor images, and is not directly related to the original feature. It can be accomplished with different visual features. We use fine-tuned R-GeM feature for the first-round retrieval and compute the affinity features for top candidates to train the re-ranking model. Then we test our model on various (type of features, fine-tuned or not) features. The performance of our method for different testing features is shown in Table 4. It obtains a large improvement relative to the baseline in several settings.

### 4.4 Comparison with the state-of-the-art methods

**mAP Comparison.** Table 5 reports the performance of our method and different compared methods on $\mathcal{R}$Oxf, $\mathcal{R}$Par, $\mathcal{R}$Oxf + $\mathcal{R}$1M, $\mathcal{R}$Par + $\mathcal{R}$1M. We compare our method with the following methods: QE and its variants as well as Database Augmentation (DBA), DSM [40], CRL [27], GSS [22], and DFS [11]. In Table 5 , if the visual feature in the first-round retrieval of comparison methods in the original paper is the same with our feature, we directly use the results in the corresponding paper. For example, the results in the second and the third blocks in Table 5 are copied from LAttQE [16]. While in the fourth block in Table 5, the visual features of comparison methods in their original paper are different from our feature. Therefore, we re-test DSM [40] and DFS [11], re-train GSS [22] with our feature by the released code for fair comparison. As for CRL [27], we re-implement it with our features by our own.

The proposed re-ranking method achieves higher performance than most of the compared methods in all settings. Our method obtains better performance compared with LAttQE [16], which is the state-of-the-art method among QE-based methods. DFS [18] and GSS [22] can achieve better performance than ours in some settings since they utilize all database images for re-ranking but inevitably incur expensive extra time costs when the database size is large. We further combine DFS and GSS with our re-ranking approach, *i.e.*, we apply our re-ranking approach to the list of retrieved results based on DFS or GSS re-ranking and the retrieval performance is further improved. Notably, when using

Table 5: mAP comparison against existing methods on the testing datasets, with Medium and Hard evaluation protocols. The performance of our method is evaluated based on the optimal settings of $K$ and $L$. AugrSfM120k and rSfM120k denote the training datasets with and without data augmentation, respectively.

| Method | Medium | | | | Hard | | | |
|---|---|---|---|---|---|---|---|---|
| | $\mathcal{R}$Oxf | $\mathcal{R}$Oxf+$\mathcal{R}$1M | $\mathcal{R}$Par | $\mathcal{R}$Par+$\mathcal{R}$1M | $\mathcal{R}$Oxf | $\mathcal{R}$Oxf+$\mathcal{R}$1M | $\mathcal{R}$Par | $\mathcal{R}$Par+$\mathcal{R}$1M |
| R-GeM(No re-ranking) [34] | 67.3 | 49.5 | 80.6 | 57.3 | 44.3 | 25.7 | 61.5 | 29.8 |
| Affinity Feature | 72.0 | 52.8 | 82.9 | 66.5 | 51.1 | 30.5 | 66.8 | 43.4 |
| AQE [9] | 72.3 | 57.3 | 82.7 | 62.3 | 49.0 | 30.5 | 65.1 | 36.5 |
| AQEwD [15] | 72.0 | 56.9 | 83.3 | 63.0 | 48.7 | 30.0 | 65.9 | 37.1 |
| DQE [1] | 72.7 | 54.5 | 83.7 | 64.2 | 48.8 | 26.3 | 66.5 | 38.0 |
| $\alpha$QE [34] | 69.3 | 52.5 | 86.9 | 66.5 | 44.5 | 26.1 | 71.7 | 41.6 |
| LAttQE [16] | 73.4 | 58.3 | 86.3 | 67.3 | 49.6 | 31.0 | 70.6 | 42.4 |
| ADBA + AQE [9] | 71.9 | 55.3 | 83.9 | 65.0 | 53.6 | 32.8 | 68.0 | 39.6 |
| ADBAwD + AQEwD [15] | 73.2 | 57.9 | 84.3 | 65.6 | 53.2 | 34.0 | 68.7 | 40.8 |
| DDBA + DQE [1] | 72.0 | 56.9 | 83.2 | 65.4 | 50.7 | 32.9 | 66.7 | 39.1 |
| $\alpha$DBA + $\alpha$QE [34] | 71.7 | 56.0 | 87.5 | 70.6 | 50.7 | 31.5 | 73.5 | 48.5 |
| LAttDBA + LAttQE [16] | 74.0 | 60.0 | 87.8 | 70.5 | 54.1 | 36.3 | 74.1 | 48.3 |
| DSM [40] | 67.1 | 50.7 | 80.5 | 57.4 | 43.7 | 26.6 | 61.1 | 30.1 |
| CRL [27] | 72.0 | 54.4 | 83.3 | 62.6 | 50.0 | 30.7 | 67.4 | 38.4 |
| GSS [22] | 78.0 | 57.8 | 88.9 | 84.8 | 60.9 | 34.3 | 76.5 | 69.2 |
| DFS [18] | 73.3 | 65.1 | 89.7 | 85.7 | 48.3 | 41.2 | 80.3 | 73.7 |
| **Ours(rSfM120k)** | 78.2 | 61.5 | 88.2 | 71.6 | 59.1 | 38.2 | 75.3 | 51.0 |
| **GSS+Ours(rSfM120k)** | 79.3 | 62.1 | 90.7 | 85.1 | 62.2 | 42.3 | 80.0 | 70.3 |
| **DFS+Ours(rSfM120k)** | 76.3 | 66.2 | 90.2 | **86.3** | 57.8 | 42.4 | 81.2 | **75.4** |
| **Ours(AugrSfM120k)** | **80.3** | 62.8 | 90.0 | 73.0 | **62.4** | 39.4 | 78.6 | 53.0 |
| **GSS+Ours(AugrSfM120k)** | 79.0 | 64.1 | **91.4** | 85.3 | 62.0 | 44.0 | 81.0 | 70.2 |
| **DFS+Ours(AugrSfM120k)** | 79.2 | **69.2** | 90.3 | 86.0 | 61.1 | **47.2** | **81.3** | 74.8 |

Table 6: Complexity comparison for different re-ranking methods. Latency is measured on an NVIDIA GTX 2080Ti GPU. $k$: number of nearest neighbors; $t$: iteration times in DFS; $K$: number of candidates to be re-ranked; $N$: number of database images; $d$: feature dimensionality; $L$:anchor image list length; Cplx: complexity.

| Method | Space Cplx. | Time Cplx. | Re-ranking latency (ms) | Memory (GB) | |
|---|---|---|---|---|---|
| | | | | $\mathcal{R}$Oxf+$\mathcal{R}$1M | $\mathcal{R}$Par+$\mathcal{R}$1M |
| DFS [18] | $\mathcal{O}(Nd + Nk)$ | $\mathcal{O}(tK^2)$ | 837 | 7.81 | 7.82 |
| GSS [22] | $\mathcal{O}(Nd + Nk)$ | $\mathcal{O}(Nd + k^2d^2)$ | 317 | 15.37 | 15.39 |
| $\alpha$QE [34] | $\mathcal{O}(Nd)$ | $\mathcal{O}(Nd + kd)$ | 184 | 7.68 | 7.69 |
| **Ours**($K$=1560) | $\mathcal{O}(Nd)$ | $\mathcal{O}(KLd + K^2)$ | 58 | 7.68 | 7.69 |
| **Ours**($K$=1280) | $\mathcal{O}(Nd)$ | $\mathcal{O}(KLd + K^2)$ | 46 | 7.68 | 7.69 |
| **Ours**($K$=1024) | $\mathcal{O}(Nd)$ | $\mathcal{O}(KLd + K^2)$ | 37 | 7.68 | 7.69 |

data augmentation, we can achieve higher performance relative to the baseline for the model trained without data augmentation. We show some successful cases and failed cases of our method on $\mathcal{R}$Oxf dataset in Figure 3.

**Speed and Memory Costs.** In Table 6, we present the complexity of the different methods in time and space as well as the measured average re-ranking latency (ms) and the total memory overhead required on the $\mathcal{R}$Oxf + $\mathcal{R}$1M, and $\mathcal{R}$Par + $\mathcal{R}$1M datasets. In terms of spatial complexity, both our method and the QE-based method [34] only involve the top-$K$ initial returned images. Differently, DFS [18] needs to encode the neighbor information between database images, resulting in an overhead to store the neighbor graph. As for GSS [34], it requires the neighbor relationship to update the features of the query image. Besides, since it involves the GCN network, the updated features cannot be used to calculate the distance directly with the original features of the database images. So additional storage of the updated features of the database images is required.

As for time complexity, DFS performs iterative operations on the top-$K$ returned candidates (generally $K = 10,000$, with over 10 iterations). QE-based methods and GSS need to first update the query features and then perform a second-round retrieval, which leads to larger latency when the size of the database becomes larger. Our method needs to firstly calculate the similarity between the top-$K$ returned candidates and the $L$ anchor images (usually $L = 512$, $K = 512$), and then use

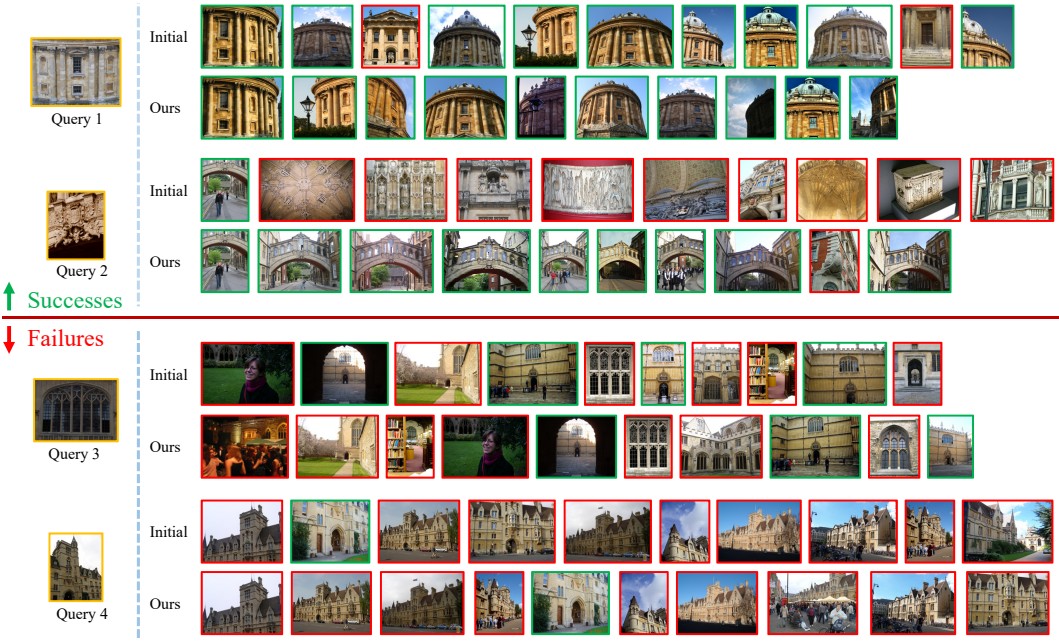

Figure 3: Selected qualitative examples of our re-ranking method. We show the top-10 results in the figure. The figure is divided into four groups each of which consists of a result of initial retrieval and a result of our re-ranking method. The first two groups are the successful cases and the other two groups are the failed cases. The images on the left with orange bounding boxes are the queries. The images with green bounding boxes denote the true positives and the red bounding boxes are false positives. Best viewed in color.

transformer encoder to generate the features, leading to a secondary complexity about $K$. Besides, the computation complexity of our method does not increase as the size of the database grows.

## 5   Conclusion

In this paper, we propose a novel visual re-ranking method for instance image retrieval. We represent the top-$K$ images in the ranking list with affinity features according to the top-$L$ anchor images. In order to explore the contextual similarity information in the ranking list, we design a self-attention re-ranking model, which updates the affinity features for re-ranking. Besides, our method is robust to the retrieval algorithm based on other image feature representation since the input of our re-ranking model is not directly related to the original image features. We further propose a data augmentation method to improve the robustness of the re-ranking model and avoid overfitting. Extensive experiments on four datasets show that our method achieves promising results compared with existing state-of-the-art methods in terms of both retrieval performance and computational time.

## 6   Acknowledgments

This work was supported in part by the National Key R&D Program of China under Contract 2018YFB1402605, in part by the National Natural Science Foundation of China under Contract 62102128, 61822208, and 62021001, and in part by the Youth Innovation Promotion Association CAS under Grant 2018497. It was also supported by the GPU cluster built by MCC Lab of Information Science and Technology Institution, USTC.

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
