# Contextual Similarity Aggregation with Self-attention for Visual Re-ranking
# <Supplementary Material>

**Jianbo Ouyang**[1]* **Hui Wu**[1]* **Min Wang**[2]† **Wengang Zhou**[1,2]† **Houqiang Li**[1,2]

[1]CAS Key Laboratory of Technology in GIPAS, EEIS Department
University of Science and Technology of China
[2]Institute of Artificial Intelligence, Hefei Comprehensive National Science Center
{ouyjb,wh241300}@mail.ustc.edu.cn
wangmin@iai.ustc.edu.cn, {zhwg,lihq}@ustc.edu.cn

## Abstract

This is the supplementary material for the paper "Contextual Similarity Aggregation with Self-attention for Visual Re-ranking" accepted to the NeurIPS 2021. This supplementary document first report the results of ablation experiments evaluated on validation set rSfM120k. Then, we analyse the limitation of our method. After that, we show the influence of different random seeds by repeating the experiment multiple times. Finally, we include the NeurIPS Paper Checklist in the supplemental material.

## 1  Ablation Experiments on Validation Set

In the manuscript, we follow the common practice in the literature to directly validate the choice of hyperparameters on the testing set. To validate the hyperparameters on the validation set to verify the merits of our choice, in this experiment, we follow HOW [6] to split the training data into a train set and a validation set. This validation set is composed of 162 3D models from rSfM120k, which is denoted as rSfM120k-HOW. This validation set is more challenging and more responsive to the target task than the original one in GeM [3]. Please refer to HOW [6] for more details. Besides, we make an additional experiment on the influence of temperature parameter. The result is shown in Figure 1. Compared with Figure 2 in the manuscript, we can find that most of the optimal parameters validated directly on ROxf (Hard) and rSfM120k-HOW are consistent.

## 2  Limitations

Our re-ranking method relies on the first-round retrieval. If the performance of the initial retrieval results is poor, our re-ranking method still works but the performance improvement is limited. However, the performance of most re-ranking methods heavily relies on the first-round retrieval, therefore this limitation is the common drawback of most re-ranking methods.

Our method just requires the affinity feature as the network input, which is generated by comparing the image with anchor images, and is not directly related to the original feature. It can be accomplished with different visual features. We use the fine-tuned R-GeM feature for the first-round retrieval and compute the affinity features for top candidates to train the re-ranking model. Then we test our model on various (type of features, fine-tuned or not) features. The result is showed in Table 4 in the

---

*The first and second authors contribute equally to this work.
†Corresponding authors: Min Wang and Wengang Zhou

35th Conference on Neural Information Processing Systems (NeurIPS 2021).

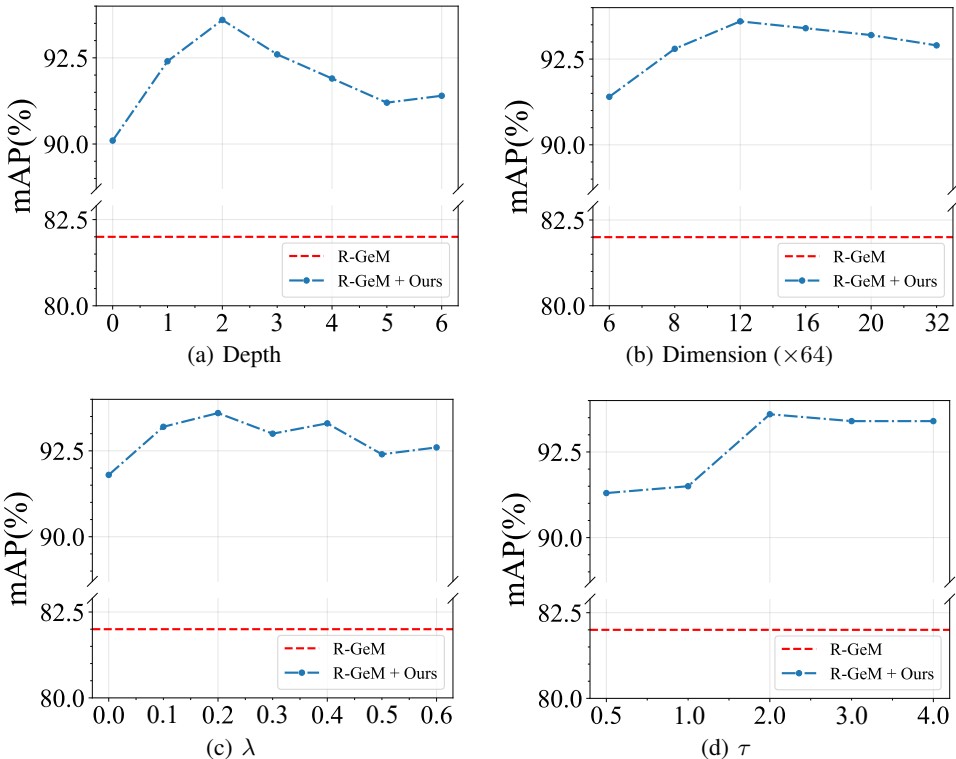

Figure 1: Impact of transformer model variants, the weight of the MSE loss, and the value of temperature on mAP on rSfM120k-HOW. All model variants are evaluated with re-ranking length $K = 1024$ and anchor image list length $L = 512$. (a) Comparisons of different transformer depth $n$. The hidden size $L' = 768$, and head number $N_h = 12$. (b) Impact of the hidden size $L'$. Transformer depth is kept as 2. (c) Impact of the weight of the MSE loss. (d) Impact of the value of temperature. The model takes the default settings.

manuscript. Our re-ranking method improves the mAP of various features with the trained re-ranking model by a large margin.

Besides, we want to confirm whether our method still works when the performance of first-round retrieval is poor, so we perform cross feature testing on two features: the off-the-shelf version of ResNet101 with R-MAC pooling and without whitening (R-RMAC[O]), the off-the-shelf version of ResNet101 with GeM pooling and without whitening (R-GeM[O]). The performance of the first-round retrieval using these two features is extremely unsatisfactory especially on the hard evaluation of $\mathcal{R}$Oxf [2] dataset. We re-rank the retrieval results of these two features with the re-ranking model trained by the fine-tuned version of R-GeM feature.

The result is shown in Table 1. The performance is enhanced by our re-ranking method for all testing features. However, the improvement is limited when using features with lower initial performance compared with the re-ranking results for features with high initial retrieval performance. We think there are two main reasons. The first point is that we only re-rank the top-$K$ candidates. When the results of the first-round retrieval are poor, there are fewer relevant images in the top-$K$ candidates. Secondly, we use affinity features as the input of our re-ranking model by calculating the similarity between the candidates and the anchor images. We directly select the top-$L$ images in the ranking list as the anchor images. When the results of the first-round retrieval are poor, the selected anchor images are far from the query, and the variance of their distances to the candidate images is relatively small, which cannot provide useful information for distinguishing the candidate images.

## 3   Impact of the random seed

The impact of different random seeds on mAP on $\mathcal{R}$Oxf and $\mathcal{R}$Par with Medium and Hard evaluation protocols is shown in Figure 2. We repeat the experiment five times. The model takes the default

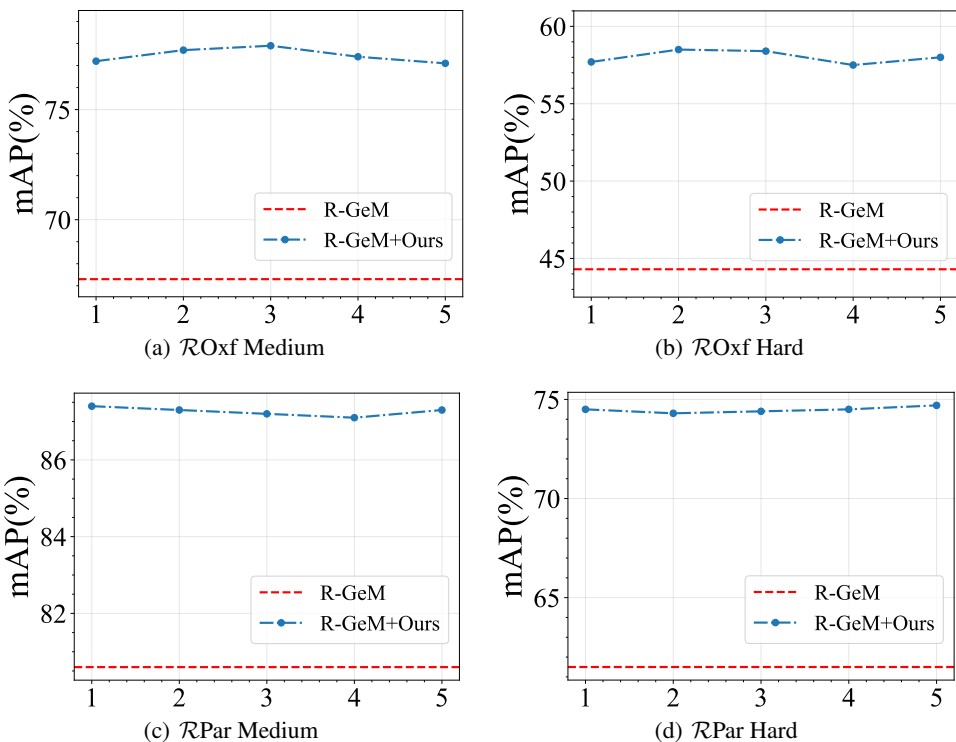

Figure 2: Impact of different random seeds on mAP on $\mathcal{R}$Oxf and $\mathcal{R}$Par with Medium and Hard evaluation protocols. The re-ranking length $K = 1024$, and the anchor image length $L = 512$. The experiment is repeated five times.

Table 1: mAP performance of the proposed model with different feature types. The re-ranking model is trained by fine-tuned R-GeM. Re-ranking length $K = 1024$. Anchor image length $L = 512$. V: VGG16 [4]; R: ResNet101 [1]; [O]: off-the-shelf networks pretrained on ImageNet; W: With post-processing whitening [3]; RMAC: regional max-pooling [5]; GeM: generalized-mean pooling [3]; MAC: max-pooling [5].

| Method | Training feature | Medium | | Hard | |
|---|---|---|---|---|---|
| | | $\mathcal{R}$Oxf | $\mathcal{R}$Par | $\mathcal{R}$Oxf | $\mathcal{R}$Par |
| R-RMAC[O] [5] | - | 26.3 | 60.0 | 5.9 | 32.6 |
| **R-RMAC[O]+Ours** | R-GeM | **29.3** | **69.7** | **9.6** | **46.3** |
| R-GeM[O] [3] | - | 21.8 | 48.0 | 5.4 | 22.3 |
| **R-GeM[O]+Ours** | R-GeM | **23.3** | **52.4** | **8.0** | **31.2** |
| R-RMAC[O]-W [5] | - | 51.2 | 74.0 | 21.4 | 51.7 |
| **R-RMAC[O]-W+Ours** | R-GeM | **56.8** | **81.9** | **30.5** | **65.7** |
| R-GeM[O]-W [3] | - | 50.3 | 73.0 | 23.0 | 50.9 |
| **R-GeM[O]-W+Ours** | R-GeM | **55.0** | **81.5** | **30.3** | **65.6** |
| R-MAC-W [5] | - | 63.3 | 76.6 | 35.7 | 55.5 |
| **R-MAC-W+Ours** | R-GeM | **73.2** | **86.0** | **52.8** | **72.1** |
| V-GeM-W [3] | - | 61.6 | 69.3 | 34.3 | 44.9 |
| **V-GeM-W+Ours** | R-GeM | **73.3** | **81.5** | **50.0** | **73.0** |

settings. The re-ranking length is set to 1024 and the anchor image length is 512. From the figure, we can see that the variance of mAP values of the five-time experiments is relatively small. Our method is virtually unaffected by the random seed and achieves stable performance on the testing datasets.