# OpenReview forum: "Contextual Similarity Aggregation with Self-attention for Visual Re-ranking"
_NeurIPS.cc/2021/Conference — NeurIPS 2021 Poster_

### Official Review · Reviewer_wuFH · 2021-07-05

**Rating:** 8
**Confidence:** 4

**Summary:**

In this paper, a visual re-ranking method is proposed. This method leverages the contextual similarity of the top-K retrieved images by comparing the original features with a set of anchor images, which is further refined by a transformer encoder. The refined features are then used to re-rank the top-K ranking list.

Moreover, a new data augmentation scheme is designed by employing different feature extractors, which enlarges the feature variety in the ranking list and improve the robustness of the re-ranking model.

Comprehensive experiments have been conducted on four benchmarks, demonstrating clear improvement and strong robustness.

**Limitations And Societal Impact:**

Limitations and potential negative societal impact of this work is discussed in the supplementary materials.

**Main Review:**

### Originality
A new re-ranking method is proposed in this paper, the proposed method differs to previous works by extracting contextual similarity from top-K images and refining them with an attention-based model. Different kinds of re-ranking methods have been included in the related work, but the discussion about the relationship between the proposed work and previous works is missing.

### Quality
The proposed method is clearly described, the contribution of each component is well demonstrated in the ablation studies. Comparison with state-of-the-art methods have also been given to support the effectiveness and generality of the proposed method. Time and memory complexity comparison is provided.

### Clarity
Overall, the paper is super clear and well structured. A few minor changes could be done to resolve the following issues:
* (**L154-156**): what is the dimension of the learnable projection matrix $W_M$, and what is the size of vector $h_i$ ?
* (**L163**): what is $S'$ and how do we initialize it ?
* (**Table-5**): what does the entry of *Affinity Feature* means? And maybe merge the second block and the third block of QE methods by having two lines for each QE such as AQE and AQE + DBA.
* (**Temperature parameter $\tau$**): seeing the importance of this parameter in the contrastive loss, maybe include an ablation study of this hyper-parameter to show how robust is the method against this hyper-param.

### Significance
The well-designed and easy-to-use re-ranking method could benefit many related applications. Besides, the usage of affinity feature and attention-based encoder could help the community understands better the effectiveness of transformer-based architecture in retrieval tasks.



**Time Spent Reviewing:**

6

---

> ### Author Response · Authors · 2021-08-09
>
> We are grateful for the insightful comments and clarify the concerns as follows.
>
> Q1：The discussion about the relationship between the proposed work and previous works is missing.
>
> A1: Thanks for your valuable suggestion. We will add a new subsection in Section 2 to discuss the relationship between the proposed work and previous works as follows:
>
> In this paper,we propose a novel visual re-ranking method by contextual similarity aggregation with self-attention.  Inspired by query expansion, we aim to update features using top-ranked image information. We use the transformer encoder to learn how to aggregate the features of the top ranked images, which is somewhat similar with LAttQE [16]. However, the input of our method is the affinity feature and is not directly related to the original visual feature, which makes our method more effective and generalizable. In addition, our method refines the features of the top-$K$ candidates rather than just refining the query feature for re-ranking.
>
> Q2: (L154-156): What is the dimension of the learnable projection matrix $W_M$, and what is the size of vector $h_i$?
>
> A2: The dimension of the learnable projection matrix $W_M$ is $L'  \times L'$, it is used to merge the information from different attention heads.
>
> The size of vector $h_i$ is $L'/N_H$, which denotes the dimension of each head in Multi-Head Attention.
>
> Q3: (L163): What is $S'$ and how to initialize it ?
>
> A3: We explain the meaning of variable $S'$ in Line 163 by Eq. (5)  between Line 163 and Line 164. In Eq. (5), $S'$ on the right denotes the output of the previous transformer encoder, while $S'$ on the left denotes the output of this encoder. At the first layer, $S'$ is initialized as the affinity matrix which is composed by the mapped affinity feature $a'_i,i=1,2,\cdots,K$ defined in Line 145. It uses a residual structure to add the output of the self-attention module to the network input, which is a common structure in transformer.
>
> Q4: (Table-5): What does the entry of Affinity Feature means? And maybe merge the second block and the third block of QE methods by having two lines for each QE such as AQE and AQE + DBA.
>
> A4: In Table 5, the entry of “Affinity Feature” means that we directly use the affinity feature computed by Eq. (1) to re-rank the top-$K$ candidates while discarding the self-attention module.
>
> Thanks for your suggestion, we will adjust the layout of Table 5 in revision.
>
> Q5: (Temperature parameter): Seeing the importance of this parameter in the contrastive loss, maybe include an ablation study of this hyper-parameter to show how robust is the method against this hyper-param.
>
> A5: We make an additional experiment on the influence of temperature parameter. The temperature parameter $\tau$ is set to the following values: 0.5, 1, 2, 3, 4. The corresponding mAP values on ROxf and RPar are shown as follows:
>
> | Temperature            | ROxf (M) | RPar (M) | ROxf (H) | RPar (H) |
> |------------------------|:--------:|:--------:|:--------:|:--------:|
> | R-GeM                  |   67.3   |   80.6   |   44.3   |   61.5   |
> | R-GeM+Ours($\tau$=0.5) |   76.1   |   86.6   |   54.2   |   72.4   |
> | R-GeM+Ours($\tau$=1.0) |   76.9   |   87.2   |   57.2   |   73.9   |
> | R-GeM+Ours($\tau$=2.0) | **77.9** | **87.2** | **58.4** | **74.4** |
> | R-GeM+Ours($\tau$=3.0) |   76.9   |   87.1   |   57.5   |   73.7   |
> | R-GeM+Ours($\tau$=4.0) |   76.8   |   86.9   |   57.0   |   73.3   |
>
>
>
>
> We can see that the model achieves the optimal performance when $\tau$=2.0, which is the setting in the manuscript. Besides, the performance is not sensitive to the temperature parameter when $\tau$ is larger than 1.

---

> > ### Comment · Reviewer_wuFH · 2021-08-29
> > **Thanks for the response**
> >
> > Thanks for the very detailed and helpful response. I'm still very positive about the paper, thus I will keep my ratings.

---

> > > ### Author Response · Authors · 2021-08-30
> > > **Thanks for your support**
> > >
> > > Thanks for your valuable support!

---

### Official Review · Reviewer_PEsN · 2021-07-15

**Rating:** 7
**Confidence:** 4

**Summary:**

The paper proposes a re-ranking algorithm for image retrieval, capable of improving the ranking results of an off-the-shelf similarity-based retrieval system, by representing each query by an affinity vector that measures similarities to a set of anchor images, and then refine this representation by aggregating it with contextual information obtained from other top-ranked retrieval results using a transformer architecture. The refined representations of the top-K images can then be used for re-ranking.

The paper shows improvements both in precision and time complexity over previous re-ranking algorithms. The time complexity improvements are mostly due to the introduction of a fixed set of anchor images, which avoids context affinity features that scale linearly with the number of top-k retrievals considered. The precision improvements are a result of a learnable context aggregation module (in the form of a transformer) and the data augmentations used for training it.

Experiments on several benchmarks show healthy improvements over prior re-ranking procedures.

**Limitations And Societal Impact:**

This paper aims to improve image retrieval (an established research area). As such, it does not introduce new negative societal impacts.

**Main Review:**

The paper introduces a re-ranking algorithm for content-based image retrieval. As mentioned above, the proposed procedure yields performance improvements both in terms of precision and computational complexity. These improvements were shown in several benchmark datasets. The proposed method is interesting as it seeks to *learn* how to best aggregate and re-rank the outputs of a similarity-based retrieval system.

I am overall positive about the paper. Nevertheless, I hope the authors can clarify/address the two following points. I will adjust my final recommendation based on that.

The paper claims to introduce a new data augmentation technique for training the re-ranking model. However, this seems to be using different CNN backbones to generate multiple features for each image, and treat these features as augmentations. This technique does improve performance significantly. However, I'm wondering if the improvements are mostly due to the use of multiple backbones, as in the case of ensembles (often used for competitions)? Is the proposed method using all backbones at test time, or only one? if only one, which one? if not, how did you ensure a fair comparison to other methods (since it would be similar to an ensemble)?

The fixed number of anchors reduces the time complexity of the proposed procedure without affecting its precision. Table 2 also shows that the method is not too sensitive to this parameter. However, I'm concerned whether this is because evaluations are being done on datasets with a relatively low number of concepts, and so a small set of anchors can cover all the concepts. Would there be a significant performance hit if the method was applied to datasets with a large number of concepts like ImageNet? Would the method need a much large number of anchors in order to maintain high performance? And if so, wouldn't the computational gains be diminished?

-----

Post rebuttal:
Thanks to the authors for the rebuttal, and the other reviewers for their insightful reviews. After carefully considering all inputs on the paper, I will increase my rating to 7. The paper makes progress in content-based image retrieval, both in terms of precision and computational complexity, by improving upon prior re-ranking algorithms. The authors also addressed my main concerns about the fairness of their comparisons. I strongly suggest the authors to clearly articulate in the paper how the algorithm operates during deployment (in order to avoid such confusions in the future).


**Time Spent Reviewing:**

4 hours split over 2 days

---

> ### Author Response · Authors · 2021-08-09
> **Response letter to reviewer PEsN**
>
> We are grateful for the insightful comments and clarify the concerns as follows.
>
> Q1: Whether the improvements are mostly due to the use of multiple backbones, as in the case of ensembles (often used for competitions)? Is the proposed method using all backbones at test time, or only one? if only one, which one? if not, how did you ensure a fair comparison to other methods (since it would be similar to an ensemble)?
>
> A1: The proposed method is not an ensemble model. During testing, we only use ResNet101-GeM feature to perform retrieval and re-rank using our model. The re-ranking setting is the same as the compared methods. The data augmentation method is only applied during training process. It generates multiple features for each training sample and just enlarges the scale of the training set, which is not involved in the testing time. Therefore, our method is not an ensemble model.
>
> Q2: Would there be a significant performance hit if the method was applied to datasets with a large number of concepts like ImageNet? Would the method need a much large number of anchors in order to maintain high performance? And if so, wouldn't the computational gains be diminished?
>
> A2: As a matter of fact, in our method for retrieval, we dynamically select anchor images from the initial rank list for each query. Different from the “anchor images” defined in common sense, in our method, the anchor images are different for different queries. Therefore, we only concern whether the anchor images are relevant to the query image, and do not care whether the anchor images cover all the concepts.
>
> On the other hand, in our paper, we conduct experiments on two large-scale datasets: ROxf + R1M and RPar + R1M. The R1M dataset is a set of 1M distractor images which are collected from Yahoo Flickr Creative Commons 100 Million (YFCC100m) dataset. It contains various classes of images and describes a variety of concepts. R1M can be regarded as a dataset with a large number of concepts. The detailed information about R1M is illustrated in [1]. As shown in Table 5, The proposed method obtains consistent promotion on ROxf + R1M and RPar + R1M. Therefore, our method can be applied to datasets with a large number of concepts.
>
> We conduct an experiment about the influence of the number of anchor images $L$ in Table 2 in the manuscript. The optimal $L$ is equal to 512 for most evaluation settings. It is because that when the number of anchor images is too large, the anchor images include too many irrelevant images, which casts a negative impact on the discrimination of the affinity feature. If the selected anchor images are far from the top-$K$ candidates in the ranking list, the variance of its distance with candidates is relatively small, which is difficult to provide useful information to distinguish the candidates. Therefore, the method needs a proper number of anchors in order to obtain high performance.
>
> [1] Filip Radenovic, Ahmet Iscen, Giorgos Tolias, Yannis Avrithis, and Ondrej Chum. Revisiting oxford and paris: Large-scale image retrieval benchmarking. In Proceedings of the IEEE Conference on Computer Vision and Pattern Recognition, 2018.

---

> ### Author Response · Authors · 2021-08-29
> **Response letter to reviewer PEsN**
>
> Thanks for your valuable suggestion and support. We will revise our manuscript and make it clarified to the readers.

---

### Official Review · Reviewer_zg23 · 2021-07-15

**Rating:** 6
**Confidence:** 4

**Summary:**

The paper introduces a new re-ranking method for image retrieval. It proposes to learn an affinity vector for each of the top-ranked candidates w.r.t a set of anchor images, using a transformer encoder. These affinity vectors are then used to re-estimate the similarity scores between the top-ranked candidates and the query image. The proposed method is evaluated on the Revisited Oxford and Paris datasets and shown to be superior or complementary to other query expansion/diffusion based reranking methods.

**Ethical Concerns:**

I don't have any ethical concerns about the paper.

**Limitations And Societal Impact:**

I don't think there will be any potential negative societal impact of the paper. For the limitations of the paper, please refer to the "concerns" part of the main review.

**Main Review:**

------------------------------------------------------------------------------------------------------------------

Strengths

[S1] The paper proposes to aggregate the contextual similarities between the query image and the top-ranked candidates for image reranking using a transformer based model. The proposed method is technically sound. Compared to other reranking methods, the proposed method is also more flexible in the sense that only the affinity vector for each image is used. Therefore, the training features and testing features can be from different feature extractors. It also enables cross-feature based data augmentation.

[S2] The paper is easy to follow.

[S3] The proposed method is shown to be superior or at least complementary to state-of-the-art query expansion/diffusion based reranking methods. Ablations on several components (e.g. # of images to be reranked, # of anchor images, data augmentation, etc) are also included.

------------------------------------------------------------------------------------------------------------------
Concerns

[C1] For the comparisons shown in Table 5, I wonder if the scores of the baselines are from their original papers or re-training/re-testing/re-implementation? As the numbers for some baselines appear to be inconsistent with those reported in their original papers. For example, the mAP for GSS[22] on RParis Medium is {92.4}/{88.9} in {the original paper}/{this paper}.

[C2] For the comparisons shown in Table 5, I wonder if the proposed method and the baselines are evaluated under similar settings. For example, i) are they using the same initial ranking lists? ii) are they using the same feature extractor, i.e. R-GeM? This is important for fair comparisons and justifying the real advantage of the proposed method.

Given the concerns listed above, I’d like to rate the paper as “marginally below the acceptance threshold” at this moment.

------------------------------------------------------------------------------------------------------------------
Minor comments

[M1] Table 5, DSM [9] → DSM [40]

-------------------------------------------------------------------------------------------------------------------
Post rebuttal

I have read the reviews and the authors' replies. In general, I think the idea of using only the affinity vectors for reranking is interesting. It makes it possible to reuse the pretrained reranking model for different feature extractors. The authors also addressed my concerns on the evaluation: if all the methods were tested under the same setting. Therefore, I raise my score and vote for accepting the paper.





**Time Spent Reviewing:**

5

---

> ### Author Response · Authors · 2021-08-09
> **Response letter to reviewer zg23**
>
> We are grateful for the insightful comments and clarify the concerns as follows.
>
> Q1: For the comparisons shown in Table 5, I wonder if the scores of the baselines are from their original papers or re-training/re-testing/re-implementation? As the numbers for some baselines appear to be inconsistent with those reported in their original papers. For example, the mAP for GSS[22] on RParis Medium is {92.4}/{88.9} in {the original paper}/{this paper}.
>
> A1: We use the ResNet101-GeM-GL18 feature which is extracted with the model finetuned on GL18 dataset to perform the first-round retrieval. In Table 5, if the visual feature of comparison methods in the original paper is the same with our feature, we directly use the scores in the corresponding paper. For example, the scores in the second and the third blocks in table 5 are copied from LAttQE [16]. While in the fourth block in Table 5, the visual features of comparison methods in their original paper are different from our feature. Therefore, we re-test DSM [9] and DFS[18], re-train GSS [22]  with our feature by the released code for fair comparison. As for CRL [27], we  re-implement it with our feature by our own.
>
> In the original paper of GSS [22], the authors exploit the ResNet101-GeM-rSfM120k feature extracted from the model finetuned on rSfM120k dataset to perform first-round retrieval, which is different from ours. With different first-round retrieval performances, the re-ranking performance is also different. Therefore the mAP we report in our paper is different from the value in the original GSS [22] paper.
>
> Q2: For the comparisons shown in Table 5, I wonder if the proposed method and the baselines are evaluated under similar settings. For example, i) are they using the same initial ranking lists? ii) are they using the same feature extractor, i.e. R-GeM? This is important for fair comparisons and justifying the real advantage of the proposed method.
>
> A2: Thanks for your valuable comments. We confirm that the proposed method and the baselines are evaluated under the same initial ranking lists and the same feature extractor (ResNet101-GeM-GL18). The experiments reported in the manuscript totally follow the protocol of fair comparison. We will revise our manuscript and make it clarified to the readers.
>
> Q3: Minor comment: Table 5, DSM [9] → DSM [40].
>
> A3: Thanks for pointing out this issue. We will carefully revise our manuscript and modify typos like this.

---

> ### Author Response · Authors · 2021-08-29
> **Response letter to reviewer zg23**
>
> Thanks for your insightful comments. We are grateful for your valuable support!

---

### Official Review · Reviewer_3tLh · 2021-07-16

**Rating:** 7
**Confidence:** 5

**Summary:**

Paper presents a method to perform top-k reranking on a (image) retrieval system. The method uses external data to learn, in a supervised manner,  a reranking module that produces contextualized representations (by means of self attention) of the top-k results, which are then used to rerank them.
The method seems inspired by the recent LattQE [16], that also uses a reranking module based on self attention and trained in a supervised manner. However, I see this paper has relevant contributions / differences:
- The proposed method uses "affinity" representations as the input of the reranking module, that probably capture better the similarity space.
- The form of supervision is different, computing a contrastive loss only amongst the top-k results.
- In addition, a "fidelity" loss and a different form of data augmentation is proposed.

**Limitations And Societal Impact:**

- Paper describes some limitations (particularly, if the first round of results is bad, reranking will not work well).

- Paper does not have a societal impact section. Although I understand this is not as obvious here as in other problems (e.g. face recognition), I think authors could have made an effort here. For example, is it possible that when trying to retrieve images with very very few relevant items (because they are not classic western-centric landmarks), the reranking actually makes the results worse?

**Main Review:**

In general, I enjoyed reading this paper.

- In terms of novelty, I think the paper is somewhat incremental from the work of [16]. However, I think enough novel technical contributions are proposed, making it, in my opinion, barely above the threshold.

- In terms of clarity, the paper is generally easy to follow, although there are some technical details I could not understand completely, I'm asking some clarifying questions to the authors below.

- The proposed method and decisions seem reasonable as well. There are some decisions I find somewhat surprising though, again, I will ask clarifying questions below.

- In terms of significance / quantitative evaluation, the method clearly outperform existing approaches by a significant margin on two standard benchmarks (plus augmented versions). The model achieves good results both on rOxford and rParis using the same model, which is not common (typically, models that work on one dataset tend to fail in the other due to differences in statistics within the datasets, for example number of relevant images per query). On the negative side, I wish the authors could have reported results on more datasets (e.g. Instre). I've seen many papers accepted reporting results only on rOxford and rParis, so this is not grounds for rejection, but I wish the authors would consider including those results as well. Also, Pytorch code was provided, which will help other researchers build on top of this work. Thank you authors for making the code available.

On the negative side, it seems to me (and please authors correct me if I'm wrong) that some hyper parameters are validated directly on test (see, e.g. Figure 2, where Roxf hard is used). This is a barely-discussed, dark standard practice in the retrieval community. I am not going to strongly penalize the authors for this when essentially every other paper gets away with it. However, I'd urge the authors to reconsider validating the hyper parameters in rSfM120k or other dataset (and perhaps add the results validating in rOxf in the supplementary material)

Questions for authors:

- It was unclear under which circumstances different features were used. Was this used only to augment the training data by generating different lists of top-k results, or was it also used to generate different affinity features (in the sense of the dot product scores being different because the features are different, not in the sense of the L anchors being different). Also, does this happen only at training time, or also at test time?

- I understand that no form of database augmentation (e.g. DBA) is happening, is that correct?

- Can you elaborate a bit more about the need for the MSE loss? It is unclear to me why we'd want the refined features to be similar to the original features (condition on an MLP transformation). Seems like an adhoc decision.

- Did you experiment training this reranking model in GL18 (since that's where the features seem to be coming from)? Not asking the authors to do this experiment if they didn't, just wondering if they actually did it and if there's any insight (e.g. performing worse because of more overfit).

**Time Spent Reviewing:**

3

---

> ### Author Response · Authors · 2021-08-09
> **Response letter to reviewer 3tLh**
>
> We are grateful for the insightful comments and clarify the concerns as follows.
>
> Q1: Supplementary experiment on other datasets such as Instre.
>
> A1: Thanks for your great support and valuable comments. As suggested, we conduct experiments on Instre dataset. Specifically, we perform the first-round retrieval by leveraging the ResNet101-GeM (R-GeM) feature which is provided by the github repository of GSS [22]. In the second round, we re-rank the top-$K$ candidates using the re-ranking model trained by rSfM120k. After performing re-ranking, the performance is clearly improved (i.e. from 69.1 to 83.0). The results of the comparison methods and ours are presented as follows:
>
> | Method                             |    |  mAP  |
> |----------------------------------- |--- |:----: |
> | R-GeM                              |    | 69.1  |
> | R-GeM + $\alpha$QE [34]              |    | 74.6  |
> | R-GeM + $\alpha$QE + $\alpha$DBA [34]  |    | 76.9  |
> | R-GeM + LAttQE [16]                    |    | 75.4  |
> | R-GeM + LAttQE + LAttDBA [16]            |    | 80.7  |
> | R-GeM + DFS [18]                     |    | 81.1  |
> | R-GeM + GSS [22]                     |    | 89.2  |
> | R-GeM + Ours                         |    | 83.0  |
> | **R-GeM + GSS + Ours**                     |    | **89.3**  |
>
>
>
> We agree that models which work well on one dataset may fail in another due to the differences in statistics (such as the number of relevant images per query) within the datasets. But in the experiments, our method still obtains satisfactory results on Instre with the model trained by rSfM120k dataset, which is attributed to two facts. First, although Instre shares different semantic information with rSfM120k, the input of our method is the affinity feature, which is not sensitive to the original visual feature. Besides, in the first-round retrieval, Instre and rSfM120k  share some overlap on the number of the relevant images in top-$K$ candidates. If the model is trained by a dataset which shares similar statistics with Instre, the re-ranking performance gain will be consistent.
>
> Q2: Validating the hyperparameters on rSfM120k or other datasets.
>
> A2: Thanks for pointing out this issue. In the manuscript, we follow the common practice in the literature to directly validate the choice of hyperparameters on the testing set. To validate the hyperparameters on the validation set to verify the merits of our choice, in this experiment, we follow HOW [1] to split the training data into a train set and a validation set. This validation set is composed of 162 3D models from rSfM120k, which is denoted as rSfM120k-HOW. This validation set is more challenging and more responsive to the target task than the original one in GeM [2]. Please refer to HOW [1] for more details. With limited rebuttal time, we have finished some  experiments on the impact of the depth of transformer encoder, hidden dimension, and the weight of the MSE loss $\lambda$, which are shown below:
>
> | Depth 	| rSfM120k-HOW[1] 	|   	|   	| Hidden dimension 	| rSfM120k-HOW[1] 	|   	|   	| Weight $\lambda$ 	| rSfM120k-HOW[1] 	|
> |:-----:	|:---------------:	|:-:	|---	|:----------------:	|:---------------:	|:-:	|---	|:----------------:	|:---------------:	|
> | R-GeM 	|       82.0      	|   	|   	|       R-GeM      	|       82.0      	|   	|   	|       R-GeM      	|       82.0      	|
> |   0   	|       90.1      	|   	|   	|   $6 \times 64$  	|       91.4      	|   	|   	|        0.0       	|       91.8      	|
> |   1   	|       92.4      	|   	|   	|   $8 \times 64$  	|       92.8      	|   	|   	|        0.1       	|       93.2      	|
> |   2   	|     **93.6**    	|   	|   	|  $12 \times 64$  	|     **93.6**    	|   	|   	|        0.2       	|     **93.6**    	|
> |   3   	|       92.6      	|   	|   	|  $16 \times 64$  	|       93.4      	|   	|   	|        0.3       	|       93.0      	|
> |   4   	|       91.9      	|   	|   	|  $20 \times 64$  	|       93.2      	|   	|   	|        0.4       	|       93.3      	|
> |   5   	|       91.2      	|   	|   	|  $32 \times 64$  	|        -        	|   	|   	|        0.5       	|       92.4      	|
> |   6   	|        -        	|   	|   	|                  	|                 	|   	|   	|        0.6       	|        -        	|
>
>
>
> Compared with Figure 2 in the manuscript， we can find that most of the optimal parameters validated directly on ROxf (Hard) and rSfM120k-HOW are consistent. We will add the results for other hyperparameters in revision.
>
> [1]	Giorgos Tolias, Tomas Jenicek, and Ondrej Chum. Learning and aggregating deep local descriptors for instance-level recognition. In Proceedings of the European Conference on Computer Vision, pages 460–477, 2020.
>
> [2]	Filip Radenovic, Giorgos Tolias, and Ondrej Chum. Fine-tuning cnn image retrieval with no human annotation. IEEE Transactions on Pattern Analysis and Machine Intelligence, 41(7):  1655–1668, 2018.
>
> Q3: It was unclear under which circumstances different features were used. Was this used only to augment the training data by generating different lists of top-$K$ results, or was it also used to generate different affinity features. Does this happen only at training time, or also at test time?
>
> A3: In the data augmentation, we first use the features extracted  from different backbones to generate different ranking lists. Then, we use the corresponding features to generate different affinity features. In other words, these features are not only used to obtain ranking lists, but also used to calculate corresponding affinity features.
>
> Besides, just as you said, it may be feasible if different features are only used to obtain ranking lists but affinity features are calculated by a single feature. We will explore this alternative in our future work.
>
> Data augmentation is applied only at training time. Training with diverse affinity features allows the network to be more robust to different first-round retrieval results (better first-round retrieval results for some features and worse for others, which might cover more test scenarios). During testing, we only use ResNet101-GeM trained on GL18 to perform retrieval and rerank using our model.
>
> Q4: No form of database augmentation (e.g. DBA) is happening, is that correct?
>
> A4: Yes, it is correct. We do not perform the database augmentation (e.g. DBA) in our experiment.
>
> Q5: Elaborate a bit more about the need for the MSE loss? Why we'd want the refined features to be similar to the original features (condition on an MLP transformation). Seems like an adhoc decision.
>
> A5: We design the MSE loss to preserve the discriminative information in the original affinity feature. The performance promotion is not obvious after adding the MSE loss according to Figure 2. The significance of MSE loss is that it improves the stability of training.
>
> Q6: Did you experiment training this reranking model in GL18?
>
> A6: Thanks for your suggestion. Since GL18 is a large-scale dataset, we find it difficult to finish all the experiments on it with limited rebuttal time. We try our best to do it, and report the performance that we have obtained as follows:
>
>
> | Method               | Training set | ROxf (M) | RPar (M) | ROxf (H) | RPar (H) |
> |----------------------|:------------:|:--------:|:--------:|:--------:|:--------:|
> | R-GeM                |       -      |   67.3   |   80.6   |   44.3   |   61.5   |
> | R-GeM+Ours($K$=512)  |   rSfM120k   |   77.0   |   85.6   |   57.0   |   71.3   |
> | R-GeM+Ours($K$=512)  |     GL18     |   78.4   |   86.7   |   58.8   |   73.5   |
> | R-GeM+Ours($K$=1024) |   rSfM120k   |   77.9   |   87.2   |   58.4   |   74.4   |
> | R-GeM+Ours($K$=1024) |     GL18     |   80.2   |   88.7   |   60.4   |   76.9   |
> | R-GeM+Ours($K$=1536) |   rSfM120k   |   78.2   |   87.5   |   59.1   |   75.3   |
> | R-GeM+Ours($K$=1536) |     GL18     | **81.2** | **89.7** | **62.0** | **78.4** |
>
>
>
> As shown in the above table, we obtain higher performance using GL18 as the training set. We believe this is because GL18 is a larger and more diverse dataset, where some images have a larger number of relevant images and others have fewer compared with rSfM120k. Using GL18 as the training set encompasses more diverse scenarios and promotes the robustness of our model.
>
> Q7: When trying to retrieve images with very few relevant items, the reranking actually makes the results worse?
>
> A7: It is possible that the re-ranking will make the results worse if the query image has very few relevant items (for example, the query has only one relevant image). In this case, the affinity feature fails to provide meaningful context to distinguish the candidates because most of the anchor images are irrelevant images.
>
> To cope with this problem, an alternative is to conduct a preprocessing procedure to judge whether a candidate retrieval list is applicable for re-ranking. We can set a similarity threshold, and if most of the similarity scores of the query with the top-ranked candidates are below the threshold, we may infer that this query has very few relevant images and we will not perform re-ranking.

---

> > ### Comment · Reviewer_3tLh · 2021-09-02
> > **Thanks**
> >
> > Thanks for the detailed response. I found your answer here and to the questions posed by other reviewers satisfying, and I'm upgrading my recommendation from 6 to 7, good paper, accept.

---

> > > ### Author Response · Authors · 2021-09-02
> > > **Thanks**
> > >
> > > Thanks for your valuable support!

---

### Decision · Program_Chairs · 2021-09-27

**Decision:**

Accept (Poster)

**Comment:**

The paper introduces a new re-ranking method for image retrieval. It proposes to learn an affinity vector for each of the top-ranked candidates w.r.t a set of anchor images, using a transformer encoder. These affinity vectors are then used to re-estimate the similarity scores between the top-ranked candidates and the query image. The reviewers found the work to be technically sound, have novelty, and produce good results. There were some concerns about the fairness of the evaluations, and many other questions of detail, which the authors have addressed effectively in their rebuttal.